# The First Case of *Fusarium falciforme* Eumycetoma in Sudan and an Extensive Literature Review about Treatment Worldwide

**DOI:** 10.3390/jof9070730

**Published:** 2023-07-06

**Authors:** Emmanuel Edwar Siddig, Ayman Ahmed, Hala Fathi Eltigani, Sahar Mubarak Bakhiet, Wendy W. J. van de Sande, Ahmed Hassan Fahal

**Affiliations:** 1The Mycetoma Research Center, University of Khartoum, Khartoum 11111, Sudan; 2ErasmusMC, Department of Medical Microbiology and Infectious Diseases, University Medical Center Rotterdam, 3000 Rotterdam, The Netherlands; 3Unit of Applied Medical Sciences, Faculty of Medical Laboratory Sciences, University of Khartoum, Khartoum 11115, Sudan; 4Swiss Tropical and Public Health Institute (Swiss TPH), Allschwil, CH-4123 Basel, Switzerland; 5Faculty of Sciences, University of Basel, CH-4003 Basel, Switzerland; 6Institute of Endemic Diseases, Faculty of Medicine, University of Khartoum, Khartoum 11111, Sudan

**Keywords:** eumycetoma, pale grains, *Fusarium* spp., recurrent disease, emerging infectious diseases, transdisciplinary one health strategy, Sudan

## Abstract

Eumycetoma is an infectious disease caused by various fungal pathogens. The disease is characterised by black and pale-yellowish grain discharge. In this communication, we report a case of eumycetoma with a pale grain foot-eumycetoma caused by *Fusarium falciforme.* The patient presented at the outpatient clinic of the Mycetoma Research Centre in Sudan. The causative agent was initially misidentified as *Aspergillus nidulans* based on its seemingly similar histopathological appearance. However, sequencing the internally transcribed spacer region of the extracted grain confirmed infection with *Fusarium falciforme*. Although the patient received Itraconazole and underwent surgical excision, the disease was recurrent. To our knowledge, this is the first report on *Fusarium falciforme* causing eumycetoma in Sudan, indicating the expansion of the geographical distribution of this pathogen. This calls for raising the awareness of healthcare providers and improving the diagnostic and surveillance systems in at-risk areas to improve the case management and reduce the threat of further spread. Considering the potential impacts of *F. falciforme* infection including threatening the global health, food security, and ecosystem balance, as well as loss of biodiversity and negative socioeconomic changes in endemic countries, we recommend the implementation of an integrated transdisciplinary One Health strategy for the prevention and control of emerging infectious diseases including *F. falciforme*.

## 1. Introduction

Mycetoma is a neglected tropical disease of serious public health concern in endemic countries in Asia, Latin America, and Africa including Sudan. This progressive destructive disease can be caused by either bacteria (actinomycetoma) or fungi (eumycetoma) infection [1]. The disease is characterised by painless subcutaneous swelling, multiple sinuses, seropurulent discharge, and the presence of grains. Additionally, the disease usually starts as a small lesion on the affected site, which is usually the foot, after the inoculation of the causative agents. However, inside the body of the host, the pathogen population clusters into granule-like structures called grains of various colours depending to the causative agents [1,2,3,4]. Therefore, the structure and colour of the grains are commonly used by healthcare providers as diagnostic features.

Eumycetoma is the most prevalent type of mycetoma in certain countries around the world including India, Senegal, and Sudan and has serious clinical courses including being challenging to treat [3]. Infection is usually acquired by introducing infectious fungi through broken skin including traumatic inoculation or through contamination of an open wound [1]. Interestingly, eumycetoma can be further subdivided into two groups according to the colour of the grains: pale-grained eumycetoma and black-grained eumycetoma. Among the black-grained eumycetoma causative agents, the most prevalent species is *Madurella mycetomatis*, followed by *Falciformispora senegalensis* and *Trematosphaeria grisea*, while the most prevalent pale-grained eumycetoma agents are *Scedosporium boydii*, *Acremonium falciforme*, *Neotestudina rosatii*, and *Fusarium* spp. [1,2,3].

There are limited cases of eumycetoma due to *Fusarium* spp. reported worldwide [5]. However, they are increasingly emerging, and infections are associated with a wide spectrum of localised diseases such as keratitis, onychomycosis, cellulitis, and mycetoma, as well as disseminated infections that are usually fatal [5]. Clinically, eumycetoma caused by *Fusarium* spp. is indistinguishable from other mycetoma infections [6,7,8,9,10,11,12,13,14,15,16,17,18,19]. Therefore, robust diagnostic techniques are needed for the early detection of this rare infection. To date, eumycetoma caused by *Fusarium* spp. has never been reported in Sudan.

In this communication, we report the first-ever case of eumycetoma caused by *Fusarium falciforme* in Sudan.

## 2. Case Report

A 40-year-old housewife from the White Nile State, central Sudan (Figure 1), was presented to the outpatient clinic at the Mycetoma Research Centre (MRC) in December 2018. The patient was complaining from recurrent painless swelling of the right foot. Her condition started three years prior to presentation with slight, painless swelling of the right foot in the medial aspect overlying the foot arch, measuring 4 cm × 5 cm in diameter. The infection had a gradual onset and progressed to affecting multiple sinuses and causing seropurulent discharge containing yellowish grains. The patient reported that she previously underwent a surgical excision under local anaesthesia elsewhere. Nevertheless, no other treatment was prescribed to the patient. She did not recall any previous history of local trauma or a family history of a similar health condition. The initial clinical examinations, medical history, and socioeconomic investigations were determined not to be related to the recent complaint; however, her geographical area of residency, White Nile State, is endemic to other mycetoma infections (Figure 1).

She had a normal pulse rate (76/min), respiratory rate (17/min), blood pressure (122/80), and temperature (37 °C). Systemic examinations including cardiovascular (CVS), respiratory, central nervous system (CNS), endocrine, and gastrointestinal (GIT) were all within the normal range. Local examination of the affected limb revealed a firm, painless, non-compressible, and non-pulsatile mass of less than 10 cm in size fixed to the deep structures and skin; moreover, the skin was normal and no hypo or hyperpigmentation was noticed. There were multiple active and healed sinuses and discharge. Nonetheless, no regional lymphadenopathy was detected.

Examining the patient liver functions showed a serum bilirubin level of 0.4 mg/dL, total protein of 7.6 g/d/L, and serum albumin level of 4.9 g/dL. Moreover, the tests revealed an alkaline phosphatase level of 91 U/L, aspartate aminotransferase (AST) level of 17 U/L, and alanine aminotransferase (ALT) level of 21 U/L. Testing renal functions of the patient revealed normal blood urea of 23 mg/dL and serum creatinine of 0.43 mg/dL. Her random blood glucose was 123 mg/dL. Furthermore, additional complete blood count examinations reported a normal total white blood cell count (WBCs) of 9.0 × 10^3^, haemoglobin count of 14.1 g/dL, and platelet count of 355 × 10^3^. Viral screening produced negative results for human immunodeficiency virus (HIV) as well as hepatitis B and C.

Moreover, ultrasound examination showed multiple superficial cavities containing fluid and echogenic thick-walled grains, which aligns with eumycetoma (Figure 2A). An X-ray of the infection site confirmed the existence of soft tissue swelling with no bone involvement (Figure 2B).

A lesional deep surgical excisional biopsy (SEB) was performed to collect a sample that was taken and split into three parts; two parts were immersed in a sterile normal saline solution to be reserved for molecular diagnosis and mycology characterization. The third part of the sample was preserved in 10% neutral buffer formalin and sent to a histopathology laboratory for further processing and examinations.

The report from the histopathology and macroscopical examination showed a normal epidermis. The lesion had ill-defined margins, with pale structures appearing at the centre of each pocket. A formalin-fixed, paraffin-embedded (FFPE) tissue block was prepared from the surgical biopsy. The tissue block was cut using a rotary microtome (Leica, Wetzlar, Germany) and 3 µm sections were subsequently obtained. The sections were stained using haematoxylin and eosin (H and E) staining. A microscopic examination performed using a light microscope on the stained sections of the sample revealed the presence of a pale grain surrounded by a zone of inflammatory cells. It also showed the existence of predominantly polymorphonuclear cells, lymphocytes, plasma cells, macrophages, and epitheloid cells (Figure 2C). However, due to this presentation, it was initially misidentified as *Aspergillus nidulans*.

Nevertheless, we attempted to culture the extracted grains to isolate the pathogen for further microbiological characterization, using the samples that were preserved in sterile normal saline. The grains were extracted from the tissue using sterile blades and washed three times in sterile normal saline. Then, the grains were cultivated using selective and non-selective media including blood agar (BA) and Sabaroud dextrose agar (SDA) supplemented with chloramphenicol (0.05 g/L). After that, the grains were incubated at a general culturing condition of 37 °C. The culture was monitored for growth every three days; however, no growth was observed after a two-month incubation period. Therefore, the grains were properly disposed of along with the culture media according to our laboratory protocol. Therefore, unfortunately, no growth was obtained. This in turn, limited our ability to evaluate and test the susceptibility of the pathogens to Itraconazole, the currently recommended treatment for eumycetoma infection in Sudan.

To confirm the causative agent, we extracted the total DNA from the grains that were obtained by SEB using the ZR Fungal/Bacterial DNA MiniPrep ™ kit (Zymo Research, Irvine, CA, USA) according to the manufacturer’s guidelines as described previously [6,7]. Amplification and sequencing of the rDNA Internal Transcribed Spacer (ITS) region were performed according to the manufacturer’s instructions [8]. To identify the causative agent at the species level, the PCR product generated with the pan-fungal ITS primers was sequenced with the BigDye Terminator v3.1 Cycle Sequencing kit on an ABI 3130 Genetic Analyzer capillary sequencer according to the manufacturer’s instructions. The sequences were analysed using Chromas software (Technelysium Pty Ltd., South Brisbane, Australia), and a Blast search was performed on the publicly available database (https://blast.ncbi.nlm.nih.gov/Blast.cgi, accessed date 13 March 2023) to identify the causative agent according to the similarity score.

The ITS sequence was 100% identical to the *Fusarium falciforme* strain CFE-139 with the accession number MN653259 and 99.81% identical to the *F. falciforme*-type strain CBS 475.67 with the accession number MG189935. Our sequences were deposited to the GenBank database under the accession number OQ200549.

Accordingly, the patient was given 200 mg of Itraconazole twice daily and 5 mg of folic acid daily. Per the treatment standard protocol at the MRC, the patient was followed-up every six weeks in the outpatient clinic to monitor the treatment progress, development of side effects, and clinical outcome. In December 2021, the patient underwent lesional wide surgical excision and was discharged with an uneventful postoperative recovery on the medical treatment. In May 2022, during a follow-up visit, the patient underwent a lesional ultrasound examination that showed no evidence of mycetoma recurrence. A second ultrasound examination was performed three months later; unfortunately, disease recurrence was confirmed. The patient was advised to continue on 200 mg Itraconazole twice daily and 5 mg folic acid daily, and they are currently undergoing clinical monitoring. However, in response to expert feedback during the revision of this report, we are currently exploring the possibility of updating the treatment plan by prescribing Voriconazole to the patient, though this depends on our ability to secure the drug, which is currently unavailable in the country.

## 3. Literature Review to Improve Our Case Management

Our extensive literature review revealed that different antifungal regimens were used to treat fusariosis. Available details about previously reported cases of Fusarium eumycetoma are summarised in Table 1.

However, azoles were still the most commonly used drugs. Itraconazole was used in 31.4% of the reported cases, followed by ketoconazole (11.4%) and amphotericin B (8.6%). Furthermore, the treatment outcome differed from patient to patient. Some patients were cured, while others developed postoperative disease recurrence. Moreover, for all the reported patients, combined antifungal therapy and surgical excision were implemented (Table 1). Nevertheless, this variation in the treatment plan could be attributed to the limitations in drug availability in the country, similar to our current situation. This underscores the urgent need for novel drugs for eumycetoma that do not require invasive surgery to reduce the potential complications that sometimes lead to disability.

## 4. Discussion

In this communication, we report the first case of a rare eumycetoma caused by *Fusarium falciforme* in Sudan. *F. falciforme* is a member of the *Nectriaceae* family and *hypocreales* order. Fungi species causing eumycetoma belong to eight different orders, with *Diaporthales*, *Eurotiales*, *Hypocreales*, *Microascales*, and *Onygenales* able to produce pale grains [1]. Interestingly, in Sudan, eumycetoma is predominantly caused by the fungus *Madurella mycetomatis*, followed by *Falciformispora senegalensis*, and both of them are characterised by the production of black grains [3].

The initial misdiagnosis of this case highlights the limitations in the current diagnostic tools for mycetoma, particularly in limited-resource settings such as Sudan. Additionally, the lack of growth in our microbial culture limited our capacity to evaluate the susceptibility of the causative agent to the locally available treatment regimes.

The identification of causative organisms at the species level is vital not only for treatment but also for understanding the disease epidemiology, the development of effective prevention and control measures, and monitoring of the effectiveness of interventions [8]. Sequencing the ITS region is currently considered the gold standard for identifying fungal species [8]. In this report, the sequence analysis presented 100% similarity with *Fusarium falciforme*. Recent advances in molecular techniques, especially those based on sequencing, show promise for the accurate and rapid diagnosis and characterization of mycetoma causative agents at the species level. Sequencing the ITS region is particularly useful when traditional techniques such as culture-based and histopathology techniques fail in identifying the causative agents at the species level [1,2,3].

Worldwide, *F. falciforme* is rarely reported as a causative agent for eumycetoma. Therefore, little is known about the epidemiology and risk factors associated with *F. falciforme* infection and transmission. Similarly, limited reports are available about other *Fusarium* species, such as *F. solani*, *F. oxysporum*, *F. subglutinans*, *F. moniliforme*, *F. keratoplasticum*, and *F. pseudensiforme*, as causative agents of eumycetoma [41,42,43,44,45].

The clinical presentation of this case was similar to other eumycetoma patients. Therefore, the patient was treated with the standard mycetoma treatment protocol, which combines medical treatment with lesional wide local surgical excision.

Resistance to the currently available antifungal agents is widely reported worldwide among the different *Fusarium* spp. including *F. falciforme* [46,47]. More than 90% of the isolates tested for Itraconazole susceptibility showed minimal inhibitory concentrations (MICs) above 64 µg/mL. *F. falciforme* was reported to be more susceptible to terbinafine, as 50% of the *F. falciforme* isolates had MICs of ≤4 µg/mL [46,47]. However, as terbinafine is unavailable in Sudan, patients have been advised to continue on Itraconazole.

Due to the globally limited investment in developing novel treatments for mycetoma, limited treatment options are currently available and the disease treatment protocols are developed mainly for the locally most common infection, which is *M. mycetomatis* in Sudan. It is currently not fully known whether eumycetoma caused by other fungal species would clinically respond to the same standard treatment. It is thus important to document the treatment outcome of patients affected by such rare causative agents. Therefore, due to its limited susceptibility to most antifungal agents, the European guidelines for fusariosis recommended using amphotericin B followed by Voriconazole to treat fusariosis. Interestingly, our reported case demonstrated a case of subcutaneous Fusarium infection treated with Itraconazole per MRC guidelines, but the patient encountered a relapse, indicating that this treatment plan is inappropriate medical therapy. Nonetheless, although an international expert has recently recommended the use of Voriconazole to treat our current patient, we could not implement this recommendation due to the unavailability of the drug in Sudan. However, considering the very dire risk of the further spread of fusariosis in the country and emergence of other similar cases in the future, we are currently working with the relevant authorities to consider the introduction of Voriconazole in the country. Nevertheless, we will also need to investigate how such an effective yet expensive drug could be made affordable for the mostly poor patients with mycetoma.

Several risk factors including climate change, globalization, and the spread of drug resistance are driving the rapid emergence and spread of novel emerging pathogens in new areas [48]. In our current study, we have identified the first-ever infection of *F. falciforme* in Sudan in the White Nile State, which has an open international border with South Sudan and hosts thousands of refugees from the neighbouring country (Figure 1). This suggests the role of high cross-country movements of forcibly displaced populations and the humanitarian respondents in the spread of infectious diseases [49,50,51]. Therefore, robust diagnostic tools including sequencing should be incorporated within the integrated disease surveillance and response systems for the early detection and accurate characterization of pathogens [8]. This will substantially improve case management as well as prevention and control strategies. Infections with *F. falciforme* have been reported among forest trees (*Acacia mangium*), industrial and medicinally important plants such as industrial hemp (*Cannabis sativa*) [52], and other economically important plants such as tobacco (*Nicotiana tabacum* L.) [53]. Additionally, *F. falciforme* infections have also been reported among edible crops including chickpea [54], onion (Allium cepa) [55], *Phaseolus vulgaris* [56], and soybean (*Glycine max* L.) [57]. Furthermore, infections with *F. falciforme* have been reported among some endangered animal species such as sea turtles [58]. These worldwide growing reports indicate that the devastating impacts of the increasingly emerging and spreading infections of *F. falciforme* are not limited to the negative health and socioeconomic effects on endemic countries in Africa, Asia, and the Americas. Additional negative impacts of the spread of *F. falciforme* include threats to food security and a loss of biodiversity, as well as endangering forest and ecosystem stability. Therefore, successful prevention and control of *F. falciforme* require the implementation of a transdisciplinary One Health strategy including the integration of disease surveillance and response systems [59,60,61,62]. Furthermore, such a strategy needs to be supported by improved diagnosis, cross-country and multisectoral coordinated surveillance systems, comprehensive documentation, and immediate public sharing of data, particularly urgently needed information such as new emergences of novel pathogens and drug resistance [59]. Moreover, more investment is urgently needed for developing novel drugs and innovative prevention and control measures [1].

## 5. Conclusions

This case report is alarming as it indicates an expansion of the geographical distribution of *F. falciforme* through novel emergence into new areas, particularly in resource-poor settings that are already challenged by limited diagnostic capacities and treatment options. Therefore, countries at risk are urgently encouraged to invest in improving their diagnostic capacity, surveillance systems, and case management as well as implementing a transdisciplinary One Health strategy for the prevention and control of emerging infectious diseases including *F. falciforme.*

## Figures and Tables

**Figure 1 jof-09-00730-f001:**
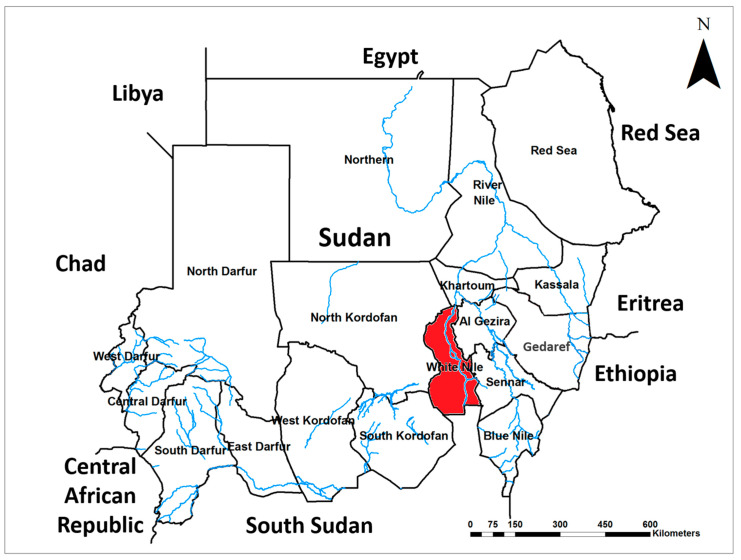
Map of Sudan indicates the original location of the first case of *Fusarium falciforme* eumycetoma in Sudan; White Nile State highlighted in red.

**Figure 2 jof-09-00730-f002:**
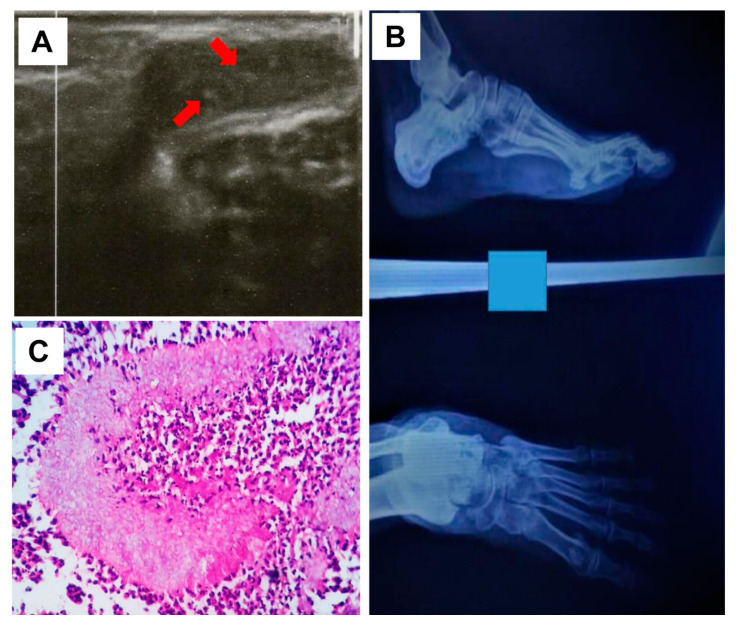
Ultrasound imaging of the infection site with multiple cavities and sharp echoes in line with mycetoma grains pointed by the red arrows (**A**), and X-ray showing the affected (left) foot confirming the lack of bone involvement (**B**). (**C**) Grains of *Fusarium falciforme* stained a pale colour with H&E and surrounded by neutrophils and epithelioid cells (H and E; 40X).

**Table 1 jof-09-00730-t001:** Geographical distribution, medical characteristics, treatment, and clinical outcome of previously reported cases of fusariosis.

Patient Origin	Site	Duration in Years	Species	Treatment	Treatment Outcome	References
Cameroon	NA	NA	*Fusarium* sp.	NA	NA	[9]
Cameroon	Foot	NA	*Fusarium* sp.	Surgery	NA	[10]
Senegal	Foot	NA	*Fusarium solani var. minus*	NA	NA	[11]
Somalia	NA	NA	*Fusarium* sp.	NA	NA	[12]
NA	NA	NA	*Fusarium solani*	NA	NA	[13]
NA	NA	NA	*F. oxysporum*	NA	NA	[13]
Jamaica	NA	NA	*Fusarium* sp.	NA	NA	[14]
India	NA	NA	*Fusarium* sp.	NA	NA	[14]
Thailand	Ankle	NA	*Fusarium solani var. coeruleum*	Antibiotics	unsatisfactory	[15]
Italy	Foot and ankle	5	*F. moniliforme*	Antibacterial antibiotics	NA	[16]
Nigeria	Foot	1	*Fusarium* sp.	Ketoconazole	cured	[17]
Surinam	Foot	NA	*Fusarium* sp.	Surgery and itraconazole	NA	[18]
Surinam	Foot	36	*F. solani*	Itraconzaole	Improved	[19]
Brazil	Hand	11	*F. solani*	Ketoconazole	Lost follow-up	[20]
France	Foot	3	*F. solani*	Itraconazole (800 mg/day)	Cured	[21]
United State of America	Hand	NA	*F. proliferatum*	Voriconazole	cured	[22]
India	Foot	0.5	*F. falciforme*	Voriconazole	Cured	[23]
Mexico	Ankle	2	*F. keratoplasticum*	Itraconazole (400 mg/day) Terbinafine (250 mg/day)	Relapse Cured	[24]
Mexico	Foot	3	*F. pseudensiforme*	Itraconazole (400 mg/day)	Lost follow-up	[25]
India	Buttock	4	*F. solani*	NA	NA	[26]
Mexico	Foot	8	*F. subglutinans*	Itraconazole	Improved	[27]
NA	Renal pelvic	NA	*Fusarium* sp.	Amphotericin B + Flucytosine and surgery	Cured	[28]
Surinam	NA	30	*Fusarium* sp.	Itraconazole and surgery	NA	[29]
NA	Ankle	7	*F. solani*	Antifungal	Improved	[30]
China	Foot	15	*F. falciforme*	Terbinafine	Improved but relapsed after one year	[31]
United State of America	Hand and Forearm	1	*F. falciforme*	Amphotericin B and ketoconazole	Diseased	[32]
India	Both feet	15	*F. solani*	Itraconazole	Improved	[33]
Romania	Foot	22	*Fusarium* sp.	Ketoconazole, penicillin, and surgery	Amputated	[34]
Cuba	Foot	18	*Fusarium* sp.	Itraconazole	Improved	[35]
Argentina	Foot	NA	*F. solani*	NA	NA	[36]
Somalia	NA	NA	*Fusarium* sp.	NA	NA	[37]
Somalia	Foot	NA	*Fusarium* sp.	Surgery	Cured	[38]
Guinea Bissau	Forearm	9	*F. solani*	Itraconazole, amphotericin B, and surgery	NA	[39]
Mexico	Foot and leg	8	*F. verticillioides*	Itraconazole and dapsone	Lost follow-up	[40]
India	Foot	NA	*F. solani*	Voriconazole and surgery	Cured	[41]

## Data Availability

All data collected during this study are included in this report.

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
