# Peer review of "The First Case of Fusarium falciforme Eumycetoma in Sudan and an Extensive Literature Review about Treatment Worldwide"

_jof, 2023, doi:10.3390/jof9070730_

Round 1

Reviewer 1 Report

There are several mistakes in the writing, as in line 21 and 33, among others.

Although it is a chronic inflammatory disease, it is an infection, and should be described as such.

Ref 3 differs to what the authors state, that eumycetoma is the most common form. Its epidemiology between actinomycetoma and eumycetoma varies geographically.

1.    What is the main question addressed by the research?

None, this is a case report

2. Do you consider the topic original or relevant in the field? Does it
address a specific gap in the field?

no
3. What does it add to the subject area compared with other published
material?

The most novel information is that it is the first case in their country
4. What specific improvements should the authors consider regarding the
methodology? What further controls should be considered?

This being a case report, methodology does not need to be modified
5. Are the conclusions consistent with the evidence and arguments presented
and do they address the main question posed?

As mentioned before, I dont think that htey can make such a conclusión about “alarming” case with just one patient
6. Are the references appropriate?

Other than the comment about one reference i already made, the others seem fine
7. Please include any additional comments on the tables and figures.

No comments

There are several mistakes in the writing, as in line 21 and 33, among others.

Although it is a chronic inflammatory disease, it is an infection, and should be described as such.

Ref 3 differs to what the authors state, that eumycetoma is the most common form. Its epidemiology between actinomycetoma and eumycetoma varies geographically.

Author Response

We are grateful to the reviewer for his/her constructive feedback that was informative and useful for improving our report. Please find below our point-by-point response to the comments

Reviewer 1:

First, we would like to thank the reviewer for their time, effort, and constructive feedback, which we believe it has helped greatly in improving our report. In addition to revising the language throughout the manuscript in response to their recommendation, below we are providing a point-by-point response to their comments and suggestions to show how and where the text was modified accordingly.

Comments and Suggestions for Authors

There are several mistakes in the writing, as in line 21 and 33, among others.

Although it is a chronic inflammatory disease, it is an infection, and should be described as such.

Response: We thank the reviewer for the valuable feedback; we have revised the language throughout the manuscript.

Ref 3 differs to what the authors state, that eumycetoma is the most common form. Its epidemiology between actinomycetoma and eumycetoma varies geographically.

Response: thank you, we revised and changed it accordingly.

  1. What is the main question addressed by the research? None, this is a case report
  2. Do you consider the topic original or relevant in the field? Does it
    address a specific gap in the field? No
  3. What does it add to the subject area compared with other published
    material? The most novel information is that it is the first case in their country

Response: The add value of this report is highlighting the expansion of this pathogen geographical distribution and the diagnostic challenge it brings into novel areas. Hopefully, this will advocate and help in improving the diagnostic capacity in areas at risk.

  1. What specific improvements should the authors consider regarding the
    methodology? What further controls should be considered? This being a case report, methodology does not need to be modified

Response: Thank you.

  1. Are the conclusions consistent with the evidence and arguments presented
    and do they address the main question posed? As mentioned before, I dont think that htey can make such a conclusión about “alarming” case with just one patient

Response: Our alarming call is related to the expansion of the geographical distribution of this pathogen instead of the number of cases and it meant to alert healthcare providers locally and the other countries in the region about the presence of this infection and the need to avoid misdiagnosis due to similarity in the clinical manifestation.

  1. Are the references appropriate?

 Other than the comment about one reference i already made, the others seem fine

Response: Thank you for raising our attention to that one, which we have corrected it accordingly.

  1. Please include any additional comments on the tables and figures.

No comments

Reviewer 2 Report

Siddig et al . Report on a rare case of Eumycetoma caused by Fusarium falciforme in a immunocompetentadult patient.

Major comment:

the case described relapsed after surgery and while on treatment with itraconazole: I suggest ot underline more the fact that the antifungal treatment of this patient could be inppropriate and give majo enphasis to the most recommended therapy for Fusarium that is Voriconazole. Nowadays, voriconazole is a generica drug, no more covered by licence so it could be affordable also in low-income countries for selected cases of difficult to treat fungal infections.

Minor:

Line 33: correct Latein-Latin

Line 141 : correct secuences – sequences

Line 150: correct asvised -advised

line 210: following ?? please delete

Line 227: correct scosystem - ecosystem

present some spelling errors

Author Response

Reviewer 2:

We are so grateful to the reviewer for their time, effort, and valuable feedback that has helped realty in improving our report. Below we are providing a point-by-point response to their comments and suggestions to show how and where the text was modified accordingly.

Major comment:

the case described relapsed after surgery and while on treatment with itraconazole: I suggest ot underline more the fact that the antifungal treatment of this patient could be inppropriate and give majo enphasis to the most recommended therapy for Fusarium that is Voriconazole. Nowadays, voriconazole is a generica drug, no more covered by licence so it could be affordable also in low-income countries for selected cases of difficult to treat fungal infections.

Response: Thank you for the extremely valuable information. We have included this recommendation in the result section and it reads “However, in response to an expert feedback during the revision of this report, we are currently exploring the possibility of updating the treatment plan by prescribing Voriconazole to the patient, though this depends on our ability to secure the currently not available in the country drug.” And also discussed it in the discussion section where it reads “Therefore, due to its limited susceptibility to most antifungal agents, the European guidelines for fusariosis recommended using amphotericin B followed by voriconazole to treat fusariosis. Interestingly, our reported case demonstrated a case of subcutaneous Fusarium infection that treated with Itraconazole per MRC guidelines but the patient encountered a relapse indicating that this  treatment plan is inappropriate medical therapy. Nonetheless, although an international expert has recently recommended the use of Voriconazole to treat our current patient, we could not implement this recommendation due to the lack of the drug in the country. However, considering the very dire risk of further spread of fusariosis in the country and emergence of other similar cases in the future, we are currently exploring with the relevant authorities the introduction of Voriconazole in the country. Nevertheless, we will also need to investigate how such effective yet expensive drug could be made affordable the mostly poor patients of mycetoma.”

Minor:

Line 33: correct Latein-Latin

Line 141 : correct secuences – sequences

Line 150: correct asvised -advised

line 210: following ?? please delete

Line 227: correct scosystem - ecosystem

Response: Thank you so much for identifying these typos, which we have corrected and revised the language throughout the manuscript in the current version.

Round 2

Reviewer 1 Report

The importance of this paper is for epidemiologic purposes, and thus, must be noted.